# Convolutional Neural Networks Adapted for Regression Tasks: Predicting the Orientation of Straight Arrows on Marked Road Pavement Using Deep Learning and Rectified Orthophotography

Calimanut-Ionut Cira [1], Alberto Díaz-Álvarez [2,*], Francisco Serradilla [2] and Miguel-Ángel Manso-Callejo [1]

1   Departamento de Ingeniería Topográfica y Cartografía, E.T.S.I. en Topografía, Geodesia y Cartografía, Universidad Politécnica de Madrid, C/Mercator 2, 28031 Madrid, Spain
2   Departamento de Sistemas Informáticos, E.T.S.I. de Sistemas Informáticos, Universidad Politécnica de Madrid, C/Alan Turing s/n, 28031 Madrid, Spain
*   Correspondence: alberto.diaz@upm.es

**Abstract:** Arrow signs found on roadway pavement are an important component of modern transportation systems. Given the rise in autonomous vehicles, public agencies are increasingly interested in accurately identifying and analysing detailed road pavement information to generate comprehensive road maps and decision support systems that can optimise traffic flow, enhance road safety, and provide complete official road cartographic support (that can be used in autonomous driving tasks). As arrow signs are a fundamental component of traffic guidance, this paper aims to present a novel deep learning-based approach to identify the orientation and direction of arrow signs on marked roadway pavements using high-resolution aerial orthoimages. The approach is based on convolutional neural network architectures (VGGNet, ResNet, Xception, and DenseNet) that are modified and adapted for regression tasks with a proposed learning structure, together with an ad hoc model, specially introduced for this task. Although the best-performing artificial neural network was based on VGGNet (VGG-19 variant), it only slightly surpassed the proposed ad hoc model in the average values of the $R^2$ score, mean squared error, and angular error by 0.005, 0.001, and 0.036, respectively, using the training set (the ad hoc model delivered an average $R^2$ score, mean squared error, and angular error of 0.9874, 0.001, and 2.516, respectively). Furthermore, the ad hoc model's predictions using the test set were the most consistent (a standard deviation of the $R^2$ score of 0.033 compared with the score of 0.042 achieved using VGG19), while being almost eight times more computationally efficient when compared with the VGG19 model (2,673,729 parameters vs VGG19's 20,321,985 parameters).

**Keywords:** convolutional neural network; regression task; road sign; pavement arrow; orientation direction

## 1. Introduction

Arrow signs on roadway pavement are a crucial component of modern transportation systems that provide critical direction and guidance for drivers. The accurate identification and analysis of these signs is important for creating comprehensive road maps and decision support systems that can optimise traffic flow and enhance road safety. Traditional methods applied to identify arrow signs on the pavement involve manual inspection, which can be time-consuming and prone to errors. However, recent advances in computer vision and deep learning (DL) can enable the automation of the process of identifying arrow signs on roadway pavement using orthophotography (it is important to note that no public dataset or repository containing road arrow signs is available).

This paper aims to present a novel approach that provides accurate and efficient identification of arrow signs on roadway pavement, together with their angle orientation and direction using aerial orthophotography and DL algorithms. The method can automatically

determine the travel direction of highways or road network lanes that flow in parallel and associate the predicted information within the scope of cartography production and updating to facilitate autonomous vehicle navigation. In this regard, the predicted information is associated with the geometries of the road axes alongside other types of details such as the number of lanes and speed limits.

Specifically, in the proposed method, convolutional neural network (CNN) architectures that were adapted for regression tasks were trained to automatically detect the orientation and direction of straight arrow signs on roadway pavement, using aerial high-resolution imagery. This approach enabled us to overcome the limitations of traditional manual inspection methods and provide a more efficient and accurate way of analysing these traffic signals. To do so, several popular CNNs (VGGNet [1], ResNet-50 [2], Xception [3], and DenseNet [4]) were adapted for regression tasks and trained with state-of-the-art techniques. This investigation was experimental and used a quantitative approach, where we raised a delimited and concrete study problem and processed data collected by applying standard processing techniques for training artificial neural networks. In the experimental design, four CNN models that have proven their effectiveness in image recognition were considered, together with an ad hoc model that was specifically designed for the task of arrow orientation recognition with the computational efficiency component in mind (being better suited for real-time applications). Afterwards, quantitative analyses and a comparison of the performance achieved in the experimental results were conducted to identify the most suitable model that can serve as a basis for a future improvement in the performance metrics or be introduced in a road extraction workflow.

The main contributions of this work are summarised as follows.

1. A deep learning-based methodology was developed that can accurately analyse straight arrow signs on road pavement using orthophotography and predict their orientation. The proposed approach was based on the adaptation of convolutional neural networks for regression tasks and was evaluated and implemented on popular deep learning image recognition models, where it achieved a maximum mean $R^2$ score of 0.993 on the training set and a maximum $R^2$ score of 0.896 on the test set.
2. A benchmark dataset (RoadArrowORIEN) was developed for predicting the orientation angle of road directional arrows, and the method applied to create it is described. The dataset can be used for training and evaluating the performance of future model implementations; it is hosted by the Zenodo repository [5] and can be downloaded under a CC-BY 4.0 licence.
3. A new artificial neural network architecture was designed to improve the performance and efficiency in the task of predicting the orientation of arrow signs found on road pavement that was specifically constructed for faster prediction times. The model achieved a mean $R^2$ score of 0.987 on the training set and a maximum $R^2$ score of 0.862 on the test set.

The remainder of this article is organised as follows. In Section 2, similar studies found in the relevant literature are discussed. Section 3 presents the proposed deep learning method. Section 4 describes the experimental design and the additional algorithmic implementations considered in this study. In Section 5, the discussion of the obtained results can be found. Lastly, Section 6 draws the conclusions of this study and mentions future lines of work.

## 2. Related Work

During the last decade, there have been significant advances in the DL field, mainly caused by the progress made in computer vision techniques—the introduced methods have impacted and affected most areas of science. In the research field related to the analysis of road pavement markings and signs, several studies have explored the use of machine learning algorithms for the identification of various road markings, such as stop lines, pedestrian crossings, and lane markings [6–8]. These studies have shown promising results in terms of accurate detection and classification of these markings.

Orthophotography was used in several studies focused on the detection of lane markings on roads. For example, Soilan et al. [9] use ortho-imagery to identify arrow signs that were manually segmented as ground truth for an application system using mobile laser scanning (MLS). Ansarnia et al. [10] use orthophotography from vertically installed cameras for pedestrian and vehicle detection, and their approach involves the use of DL for different tasks including image classification (where the YOLO algorithm [11] was used). Both papers discuss the potential for DL-based approaches to accurately detect the position of elements on the road transport network. In addition, Pritt et al. [12] use satellite orthophotography and DL techniques for the identification of traffic objects, thus overcoming the existing limitations of traditional object detection and classification. Specifically, this approach made use of an ensemble of deep CNNs for object recognition in high-resolution, multispectral satellite images.

As Malik and Siddiqi [13] also indicate (who propose a feature point detection and description algorithm with scale invariance and rotation invariance algorithm called BRISK), existing approaches for traffic signal extraction (in particular, vertical signals) apply more classical techniques, such as the scale-invariant feature transform (SIFT) algorithm [14], to detect and describe local features in digital images, and the Speeded-Up Robust Features (SURF) [15] computer vision algorithm, to obtain a visual representation of an image and extract detailed and content-specific information.

Li et al. [16] detect traffic signals over real-time video with YOLO-V4-tiny and YOLO-MobileNet networks, while Zhou et al. [17] use an improved version of VGG (IVGG) to detect traffic signals in Germany. Other works identified in the survey carried out by Sanyal et al. [18] (where different databases are used to test the algorithms for traffic signs in real-time video) apply different classifiers such as support vector machine [19], Gaussian, multilayer perceptrons, and convolutional neural networks that feature max pooling and fully connected layers.

In the field of object orientation detection, works that extend beyond the last decade can be found. Rybski et al. [20] determine the global orientation of vehicle trajectories from images by training an ensemble of histogram of oriented gradient (HOG) classifiers and counting instances of gradient orientation in localised parts of an image. Asad and Slabaugh [21] use random forest [22] to detect angles in hand positions registered with images, while Sun et al. [23] propose the BiFA-YOLO model as a bidirectional feature fusion and angular classification architecture based on YOLO to detect ship orientation on high-resolution synthetic aperture radar (SAR) images.

Shi et al. [24] propose an object detection method for remote sensing images that is based on angle classification and uses rotation detection bounding boxes labelled with angle information. Specifically, they incorporate the neural architecture search framework with a feature pyramid network module (NAS-FPN) in a dense detector (RetinaNet) and use a binary encoding method in angle classification. Zhao et al. [25] propose a modification of the YoloV5 framework to detect the orientation of the bounding boxes of objects and apply it in the field of electrical insulators on electricity transmission towers.

In a more recent study, Yang and Yan [26] propose the transformation of the regression problem into a circular classification problem (CSL), for which they develop an object heading detection module that can be useful when exact heading orientation information is needed (e.g., for detecting the orientation of ships and aeroplanes). Also, Wang et al. [27] evidence that using CSL does not work well because of the type of loss function used and propose the use of classification loss with adaptive Gaussian attenuation on the negative locations to solve the problem of negative angles and achieve better accuracies in angle estimation.

Finally, Zhao et al. [28] propose a robust orientation detector (OrtDet) to solve the object angle problem, since convolutional neural networks do not explicitly model orientation variation. For this purpose, the authors use the token concatenation layer (TCL) strategy, which generates a pyramidal hierarchy of features to address different scales of objects and define the mean rotational precision (mRP) as a performance metric.

The mentioned studies demonstrate the potential for DL approaches in the analysis of road markings and road signs, but they tend to focus on the identification of individual elements in very favourable remote sensing scenes. Therefore, the closest identified studies (described in this section) generally use YOLO-based networks to identify the orientation of the enveloping rectangle of the objects (and allow the recovery of the object), but not the arrow direction. This also implies that these systems are not capable of differentiating the direction of arrows found in parallel highway lanes oriented in opposite directions.

It is important to note that no methodological proposal was found in the literature to identify the orientation of a traffic direction arrow in roadways and no studies that analyse the angle of arrow signs on road pavement were identified (although this source of information is important for the identification, construction, and updating maps of the road transport network and road intelligence systems). For these reasons, this study presents a novel approach for the analysis of directional arrow signs on road pavement using orthophotography and DL techniques.

## 3. Method Proposal

The process can be divided into two phases: the dataset generation step and the comparative study of methods for the model selection step.

The first part of the process is described in Sections 3.1 and 4.1 and concerns the creation of a custom dataset for the considered task (the detection of arrow orientation in orthophotos). To obtain the data, a fine-tuned YOLOv5 algorithm (introduced by Redmon et al. [11] and modified by Jocher et al. [29]) is first used to detect and extract arrows from the original orthoimages. Afterwards, for each arrow, the rotation angle is identified with the process explained in Section 3.1. However, it was observed that the arrow recognition and orientation processes may produce inaccuracies that can be categorised into two types: (1) arrows with correct angles but opposite directions (rotated by 180 degrees) and (2) arrows that are undetectable due to potential shortcoming of the YOLO process. For the first type of error, manual corrections are applied to adjust the rotation angle, while for the second type of error, the indetectable arrows are removed from the dataset.

The second part of the process begins with the proposal of a learning structure that enables convolutional neural networks to be used in regression tasks (where the goal is the prediction of continuous values instead of class probabilities—as described in Section 3.2). Afterwards, the generated arrow signs dataset is used to train a range of popular CNN models that were modified with the proposed adaptation for regression tasks, along with an ad hoc model (described in Sections 3.2 and 4.2). For training, a cross-validation approach is applied by creating ten random partitions of the dataset. Each combination of partition and model architecture is trained independently, and the performance metrics are calculated for each partition and recorded for further analysis (as described in Sections 3.3 and 4.3).

To provide a robust assessment of model performance, a statistical analysis is performed using the bootstrap method. This enables the calculation of mean and confidence intervals for each metric, providing a comprehensive view of the model's performance. Finally, a comparative study on the performance of the considered models is carried out to identify and select the most suitable one for the task. The process described above is presented in Figure 1.

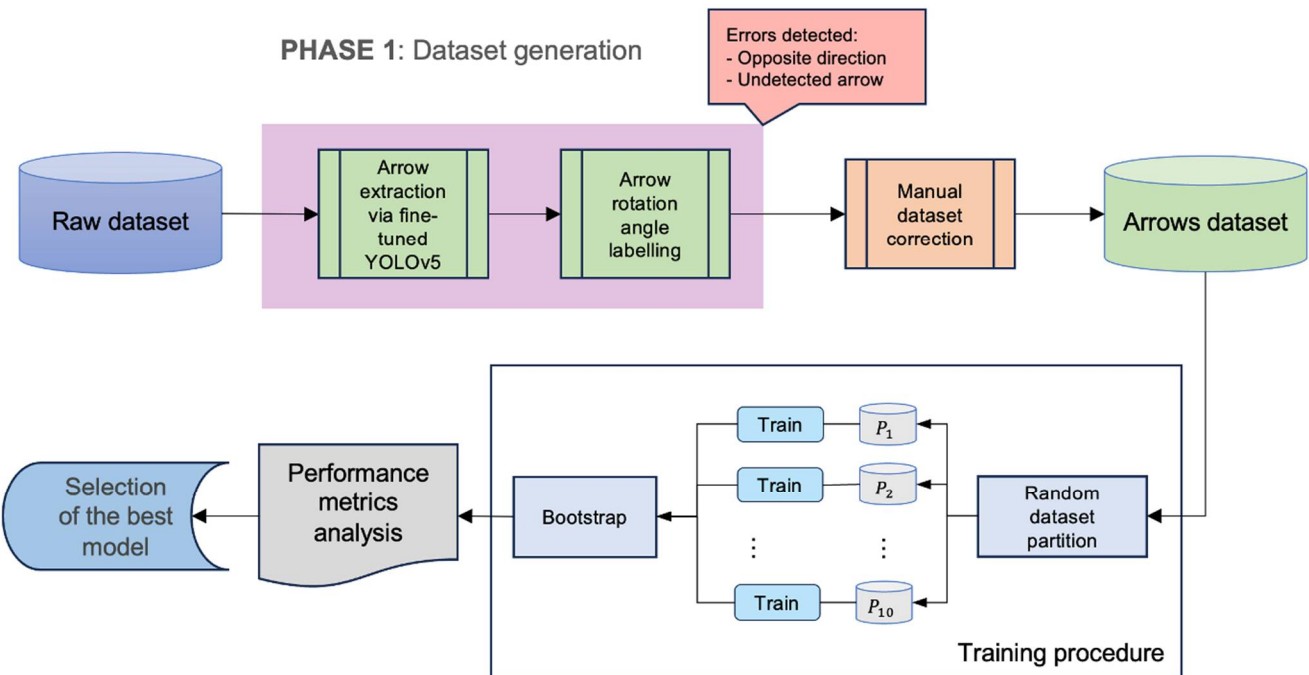

**Figure 1.** Process diagram showing the workflow applied in this study that includes the generation of an arrow dataset from orthophotography and the evaluation method to determine the final selected model.

### 3.1. Data Generation Procedure: Traffic Lane Arrow Direction and Heading Detection

The procedure for inferring the angles is based on arrow data labelled at the pixel level and includes an algorithm created to obtain the coordinates of the vertices of the polygon and perform a clustering of the points into two groups based on proximity. During labelling, each arrow was represented as a polygon and features two points at the origin and five points in the part that marks the orientation.

In the first part of the procedure, these points were processed to perform a clustering operation based on the distance between points, in such a way that from these, two clusters, $Cl_1$ and $Cl_2$, that contain two and five points, respectively, were generated. The result of applying the clustering was two classes of points, one with more points (the part of the arrow) and another with only two points (the centroids of the clusters). Afterwards, the centroid of both clusters was calculated using the K-means algorithm [30], allowing for two labelled centroid points, where one was the origin of the vector while the other was the end. The orientation angle was calculated as the azimuth between $Cl_1$ (the arrow origin) and $Cl_2$ (the arrow end). The azimuth of the vector formed between the origin and the tip, i.e., the angle with respect to the Y-axis, was calculated and afterwards used to label the images. Finally, a sub-image centred on the arrow was extracted from the tile to work with images that only contain one arrow while maintaining the angle label.

The procedure applied for generating the dataset is presented in Figure 2 and described as follows.

1. From the input consisting of RGB (red, green, blue) orthoimages, manually labelled with arrow sign information, create a JSON (JavaScript Object Notation) file containing the arrow polygon using the capabilities of software specialised in image tagging.
2. Extract the vertices of the generated arrow-shaped polygon.
3. Generate two clusters of nearby vertices, with a minimum cluster size of two vertices, so the origin cluster ($Cl_1$, containing fewer vertices) and the arrow cluster ($Cl_2$, containing five vertices) are identified.

4. For the two generated clusters, obtain their centroid ($Ce_1$ and $Ce_2$, respectively), preserving the information on the number of vertices that define the cluster.
5. Afterwards, generate the vector with origin in $Ce_1$ (of the cluster with fewer vertices) and with the end in the $Ce_2$ centroid (of the cluster with the higher number of vertices).
6. Next, calculate the azimuth of this vector with respect to the ordinate axis. For the output, automatically crop the orthoimage with a constant size (for example, $64 \times 64$ pixels) by taking an extension slightly larger than the area occupied by the arrow in the scene.

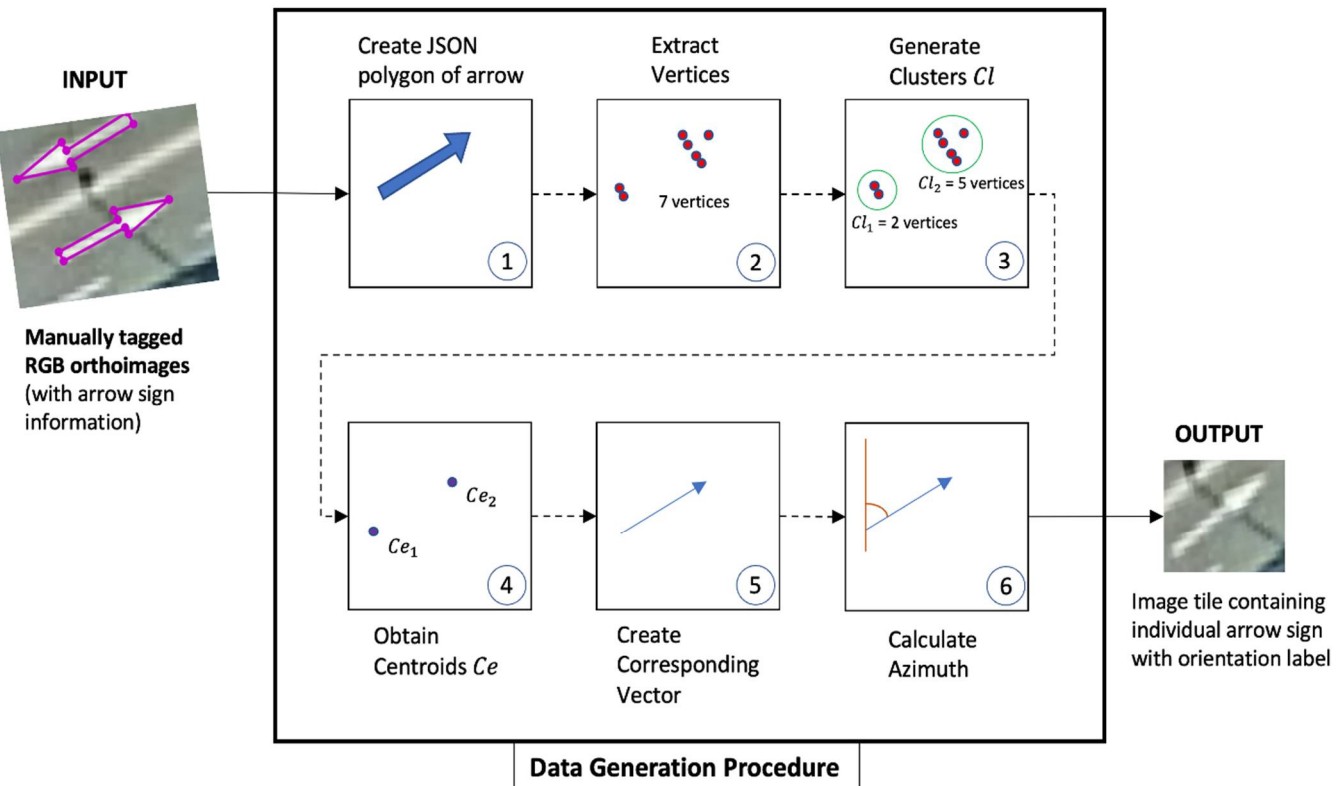

**Figure 2.** Proposed procedure for generating the dataset containing arrow signs found on pavement and their corresponding orientation label.

### 3.2. Proposed CNN Adaptation for Regression Tasks and Ad Hoc Model Architecture

As stated in the Introduction, and described in Section 3, this work aims at implementing a deep learning-based approach to predict the orientation of straight arrows on marked road pavement.

At its core, a CNN is formed by a feature learning part (or convolutional base), where convolutional and pooling layers are used to learn and extract characteristics from the available data that enable correct predictions. Afterwards, the classifier part (generally formed by fully connected, or FC, layers) is found, where the filters containing the representations learned are used for class prediction. It is important to mention that the classifier part of convolutional neural networks features fully connected layers with thousands of units and is generally prepared for image recognition challenges on large datasets (for example, many of the popular CNNs were developed to participate in the ImageNet Large Scale Visual Recognition Challenge (ILSVRC) [31], where the proposal of better learning structures was incentivised to better predict the 1000 classes featured in the ImageNet dataset that contains more than 1.2 million images).

The adaptation of CNNs for the regression task (presented in Figure 3) involves removing the classifier part of a CNN architecture and replacing it with a flatten layer and four different dense layers with 512, 64, 32, and 1 unit, respectively. It is important to note

that the final layer features a sigmoid activation function to make it suitable for regression problems. In addition, to strongly reduce the overfitting behaviour, the regression structure also features a dropout layer between the flatten and FC layers, with a rate of 0.5 (to randomly set 50% of the units to zero in each training iteration). This distribution of layers represents the inference block of the orientation angle and enables the CNN architectures, originally designed for image classification tasks, to be used in regression tasks (i.e., in this study, the target value is the angle in degrees relative to the azimuth). This architecture pivot enables the CNNs, initially architected for image classification, to be repurposed for regression problems.

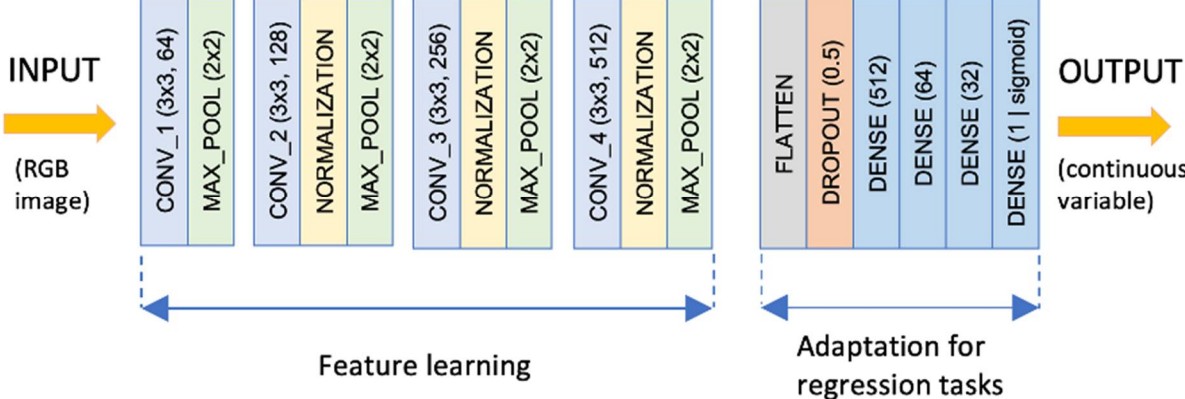

**Figure 3.** The proposed ad hoc architecture is based on CNN learning structures together with the proposed CNN adaptation for regression tasks.

Unlike expansive CNN architectures common in the literature, the ad hoc model champions simplicity without sacrificing performance. The design intent was two-fold: (a) efficiently predict arrow orientations and (b) ensure compatibility with real-time applications. The novel ad hoc architecture described in Figure 3 is designed to balance the need for feature extraction with computational efficiency and is intended to be used in a real-time application.

The ad hoc model can be seen as a CNN-based architecture with a simpler disposition of layers when compared with popular models existent in the literature (as described in Section 4.2). The architecture consists of four convolutional blocks featuring a kernel size of $3 \times 3$ with ReLU [32] activation (chosen for its computational efficiency and adeptness at introducing non-linearity, which is used after each convolution) to process the $64 \times 64 \times 3$ RGB image tensor. The four distinct blocks act as the backbone of this model and process the input image tensor, extracting intricate patterns essential for the regression task. Each convolutional block ends with a max pooling layer over a $2 \times 2$ window, ensuring a dimensionality reduction without information loss. Starting with the second convolutional block, the ad hoc model features normalisation layers to standardise the input values across the learned features within the same range to ensure more stable training and a maintain consistent data distribution across learnt features.

In the convolutional blocks, the ad hoc model applied the escalating filter count strategy, and the number of filters per convolution increases (from 64 to 128, 256, and 512) across blocks to ensure an optimal balance between basic and advanced feature extraction. The progression of these blocks—from basic to advanced feature extraction—is deliberate, mirroring the complexity of the features they are designed to capture.

Regarding efficiency and efficacy, the ad hoc architecture is fine-tuned for both feature extraction prowess and computational agility. A testament to its streamlined design, the model boasts a mere 2,673,729 parameters—a stark contrast to traditionally bulky CNNs, yet without a compromise in performance.

### 3.3. Considerations Regarding the Training Procedure

To reliably estimate the error achieved using each model, the training is repeated $N$ times (in our case, $N = 10$) with different random partitions in the train/ test data. This way, $N$ estimates of the metrics (mean squared error, $R^2$, etc.) are obtained, and the bootstrap technique [33] is applied afterwards to determine a confidence interval for each metric, without having to assume a normal distribution.

Once the $N$ estimates for the metric of interest are obtained, the statistical estimator (e.g., the mean) is calculated at a 95% confidence interval. Here, the bootstrap procedure is applied, which roughly resamples the results obtained $M$ times (in our case, $M = 10,000$) and calculates the estimator for each resampling. By sorting and eliminating the 2.5% of the values (in our case, 250) at each tail of the sorted list, the confidence interval for the estimator is obtained. Figure 4 shows the distribution of bootstrapped $R^2$ values for one of the trained models (a modified VGG19 network).

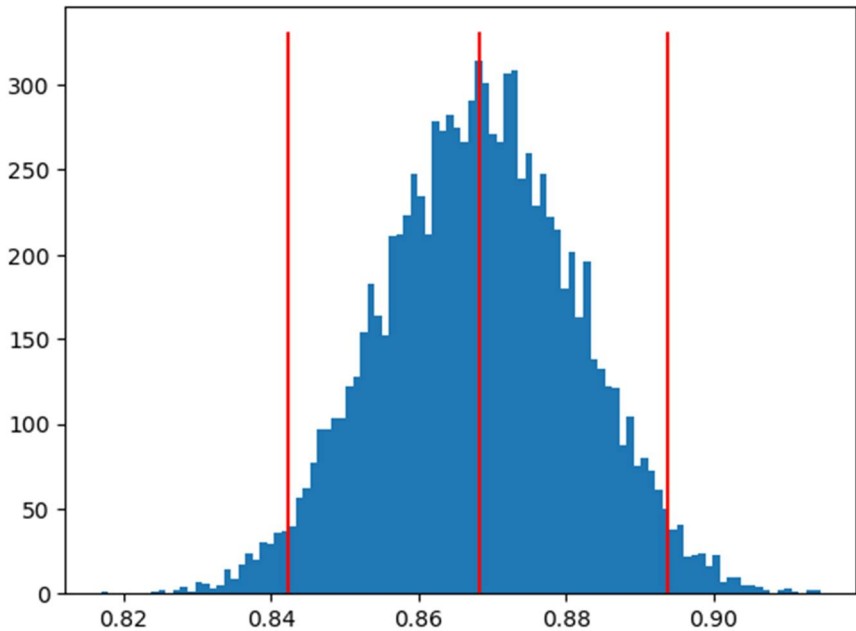

**Figure 4.** Example showing the distribution of bootstrapped $R^2$ values for one of the neural networks trained in this study (the modified VGG19 model).

Another important training aspect is that, although the data augmentation technique was proven to increase the generalisation capability of the models when the size of the training set is reduced (features less than 10,000 samples), it is fundamental not to apply data augmentation in the form of random height or width shifts, vertical and horizontal shifts, or random rotations to the image tensor in similar studies. Nonetheless, data augmentation parameters such as changes in brightness and contrast or shifts in gamma and channel intensities could help in exposing the model to more aspects of the data (if small parameter values are selected). In addition, the use of transfer learning for the convolutional base of the considered CNN networks is recommended to take advantage of their learned feature extraction capabilities.

## 4. Implementation of the Proposed Method

In this section, the implementation of the method proposed is presented. First, the dataset is generated by applying the process presented in Section 3.1. Afterwards, the popular convolutional neural networks considered in this study (for comparison with the ad hoc model) are described, and the experimental design is presented.

### 4.1. Data

The dataset used for training and testing the algorithmic implementations includes 6700 images containing arrow signals found on road pavement. The data were obtained by analysing satellite orthophotos produced by the Geographical National Institute of Spain (National Plan of Aerial Orthophotography, or PNOA product [34]) using a YOLOv5 algorithm that was fine-tuned for the task of road arrow symbol recognition.

PNOA provides digital aerial orthophotographs of the entire Spanish territory at a spatial resolution of 25 cm. The images are obtained every two to four years and are typically acquired during the summer when lighting conditions are consistent. The orthophotos used were previously radiometrically balanced and homogenised and have corrections applied to minimise the topographic and atmospheric effects. The images also feature geometric corrections aimed at eliminating distortions caused by the geometry of the sensors.

Each input in the dataset consists of an aerial image of the road pavement that contains a directional arrow. The arrow images were initially labelled as polygon-shaped arrows using LabelMe [35]. During the labelling and revision process, the quality of each arrow was checked and, if required, specific actions were taken, such as deletion of the sample if no arrow existed in the image or the direction of the arrow was unclear as well as the rotation of the arrow angle 180 degrees if the labelled angle corresponded to the opposite direction. The resulting dataset contains 6701 images of 64 × 64 pixels (examples can be found in Figure 5), together with their corresponding azimuth as the label (orientation angle in sexagesimal degrees).

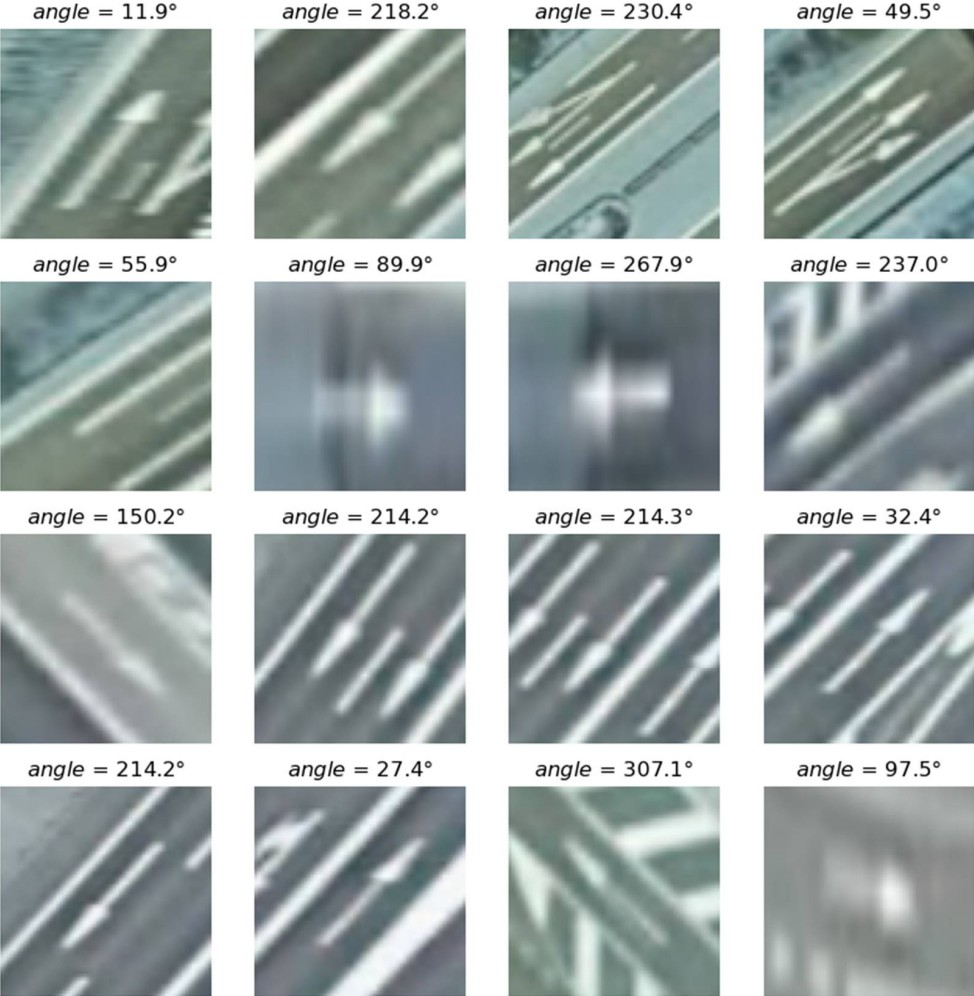

**Figure 5.** Examples of tiles belonging to the RoadArrowORIEN dataset (used for training and testing the artificial neural networks), together with their corresponding angular value.

The expected output for each input image was the rotation angle of the arrow, and the manual review process described above was essential to ensure the accuracy and consistency of the dataset. This dataset is expected to provide a significant benchmark for evaluating the performance of the different models developed for the orientation recognition of road directional arrows.

### 4.2. Popular Convolutional Neural Network Architectures Considered

The base networks selected for this study are convolutional neural networks, as other types of networks are not as suitable for extracting features as intended (for example, the YOLO model extracts the rotated rectangle that best fits the object [36]).

In addition to the ad hoc architecture described in Section 3.2, we opted for implementing several other architectures from the area of image recognition, namely, VGGNet [1] (VGG16 and VGG19 variants), ResNet-50 [2], and Xception [3], proposed for its computational efficiency. In this regard, VGGNet-based variants have demonstrated their efficiency in image recognition tasks and are widely used in the specialised literature, whereas Xception and ResNet-50 feature a more complex structure that enables a better extraction of complex features from images. It is important to mention that, for training, all the additional neural networks presented in this section were adapted for the regression task, following the CNN adaptation for the regression task proposal from Section 3.2.

### 4.2.1. VGGNet

The VGG16 and VGG19 variants of VGGNet [1], illustrated in Figure 6, are well-known, popular CNN models for image classification. The feature learning part of both networks consists of several convolutional layers containing $3 \times 3$ convolutional filters with stride and padding of size one, followed by max-pooling layers with stride of size two (for the feature learning part). The main difference between the two architectures is the number of layers, VGG19 features 19 layers in the feature learning part, three more than VGG16.

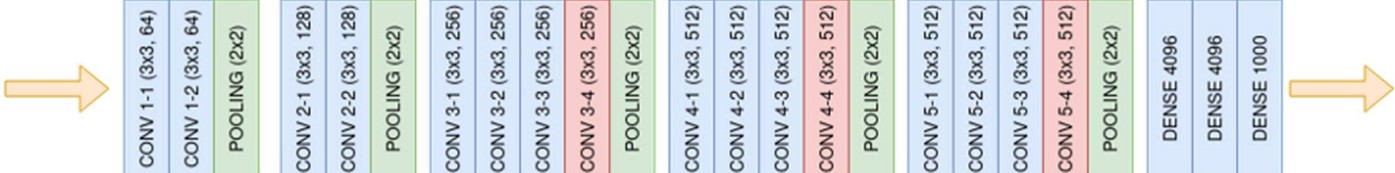

**Figure 6.** Illustration showing the VGG-16 and VGG-19 architectures. Note: VGG-16 is equivalent to VGG-19 but without the "CONV_3-4", "CONV_4-4", and "CONV_5-4" layers. Note: for training, the classifier part (the last three FC layers) was replaced with the inference block of the orientation angle proposed in Section 3.2.

In the classifier part, at the end of VGGNet (and its VGG16 and VGG19 variants), two fully connected (FC) layers with 4096 units, together with a final FC layer containing 1000 neurons, can be found (corresponding to the number of classes in the ImageNet dataset [31]).

### 4.2.2. ResNet-50

ResNet-50 (shown in Figure 7) is a residual neural network that was introduced by He et al. [2] in 2016. The main idea behind ResNet-50 is the use of residual blocks, which allow for the training of very deep neural networks by addressing the problem of vanishing gradients. This is achieved by adding skip connections that bypass one or more layers and allow the gradient to be propagated more easily through the network.

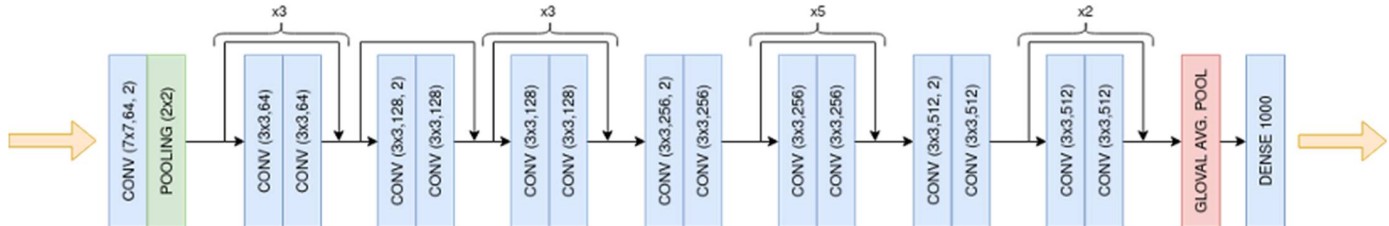

**Figure 7.** The ResNet-50 architecture consists of 50 layers, including convolutional layers, pooling layers, residual blocks, and a global average pooling layer. Note: for training, the last FC layer was replaced with the inference block of the orientation angle proposed in Section 3.2.

### 4.2.3. Xception

Xception (presented in Figure 8) was introduced in 2015 by Francois Chollet [3] and is a variant of the Inception [37] model based on separable depth-wise convolutions, which achieves a significant reduction in computational cost while maintaining the accuracy of the model. Different from VGGNet, ResNet-50 (presented in Section 4.2.2) and Xception feature a single FC layer with 1000 units (the number of output classes of the ImageNet challenge [31]).

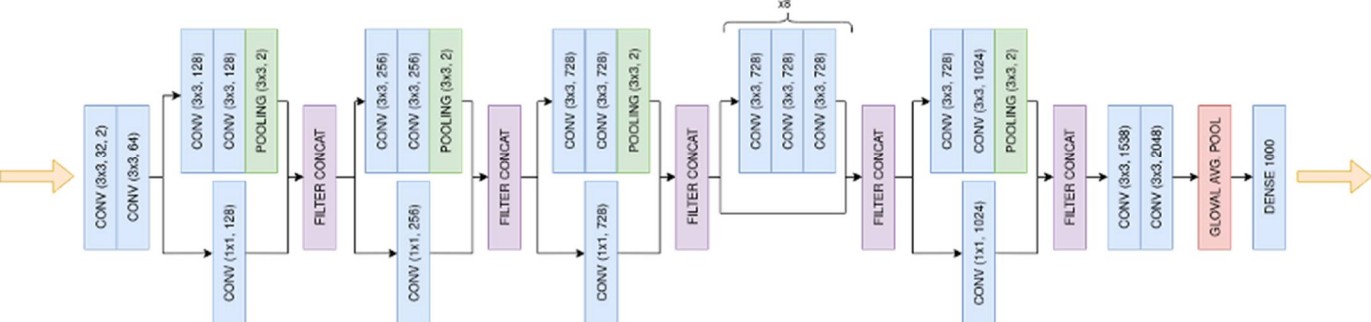

**Figure 8.** The Xception architecture consists of a series of convolutional and depth-wise separable convolutional layers, with skip connections, batch normalisation, and global average pooling. Note: for training, the last FC layer was replaced with the inference block of the orientation angle proposed in Section 3.2.

### 4.2.4. DenseNet

DenseNet (presented in Figure 9) was introduced in 2016 by Huang et al. [4] and presents a paradigm shift in the construction of CNNs. Unlike the sequential arrangement of layers found in architectures such as VGGNet, ResNet-50, and Xception (elaborated in Sections 4.2.1–4.2.3), DenseNet exhibits a dense connectivity feature, where each layer in a DenseNet block receives inputs from all preceding layers and passes on its own feature-maps to all subsequent layers. These dense connectivity patterns promote feature reuse and significantly reduce the computational burden while maintaining or even enhancing the accuracy of the model. DenseNet can be viewed as a CNN that densely connects layers featuring the same size of feature maps through the dense block structure to enable the input of additional information from previous layers while passing the learned feature maps to subsequent layers found within the same dense block. Similar to its counterparts, DenseNet features a single FC layer with 1000 units, corresponding to the number of output classes in the ImageNet challenge.

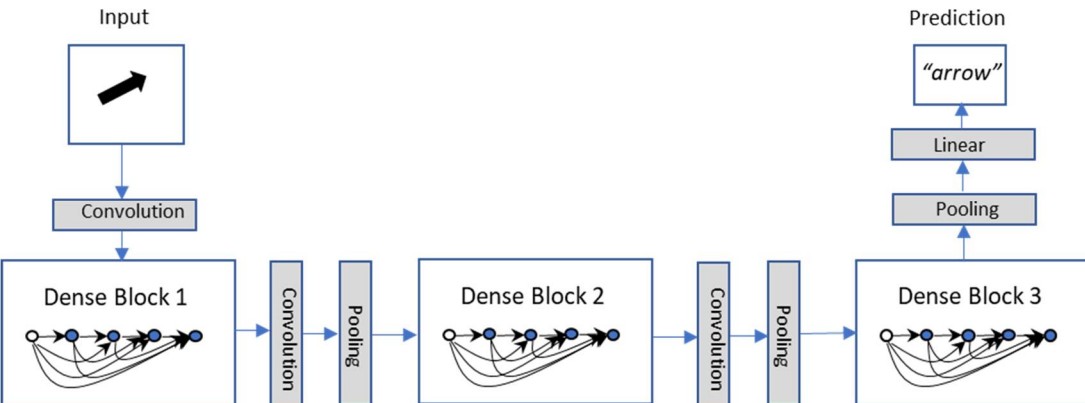

**Figure 9.** A schematic representation showing the DenseNet architecture, illustrating the information flow through the densely connected layers (based on [4]). Between the three dense blocks, two adjacent blocks are referred to as transition layers, which change feature-map sizes via convolutional and pooling layers. Note: For training, the classifier part was replaced with the inference block of the orientation angle proposed in Section 3.2.

### 4.3. Training Experiments

The considered ANN architectures were trained using the dataset described in Section 4.1. The experiments were carried out using a MacBook Pro M1 Max with a 12-core CPU (central processing unit), a 38-core GPU (graphics processing unit) with a 16-core Neural Engine, 32 GB (gigabytes) of unified memory, and 1 TB (terabyte) of SSD (solid-state drive) storage in TensorFlow [38], installed within a Python environment.

The dataset was randomly split into training and test sets by applying a 90:10% division criterion. As explained in Section 3.3 (and illustrated in Figure 1), the random division of the dataset for training and validation involved bootstrapping in the training so that the division of the dataset and the training/validation were repeated ten times to reduce the variance and avoid overfitting. This training approach, applied consistently to all the considered models, involved optimising the mean squared error (MSE) loss function, defined in Equation (1) (where each predicted value ($\hat{y}_i$) was subtracted from the actual target value ($y_i$), the differences were squared, the mean of the resulting error array was the loss to be optimised), using Adam [39] with a learning rate of 0.0001 and a batch size of 512.

$$MSE\_loss = \frac{1}{n} \sum_{i=1}^{n} (\hat{y}_i - y_i)^2 \qquad (1)$$

The models based on popular CNNs, as described in Section 4.2, were trained with transfer learning until convergence was achieved or until 50 epochs were completed. When there was no improvement in the loss using the training dataset for the past ten epochs, it was considered that the point of convergence was reached. The ad hoc model proposed in Section 3.2 was trained for 500 epochs since it has the disadvantage of starting learning from scratch.

## 5. Results and Discussion

To evaluate the effectiveness of the five trained models, a comprehensive set of evaluation metrics was utilised, including the loss value, the $R^2$ score (defined in Equation (2), where $SS$ represents the sum of squares, with $SS_{res}$ tending to a minimum, $\hat{y}_i$ represents the predicted $y$, and $\overline{y}$ is the average of the values, and $n$ is the sample size), and the mean angular error (defined as the sum of the angle errors divided by the total number of samples).

$$R^2\ score = 1 - \frac{SS_{res}}{SS_{total}} = 1 - \frac{SUM(y_i - \hat{y}_i)^2}{SUM(y_i - \overline{y})^2} \qquad (2)$$

Moreover, the consistency of the models in predicting the target variable was analysed by investigating the standard deviation in the test $R^2$ score of each model (defined in Equation (3), where $x_i$ represents any $R^2$ score value, $\overline{x}$ is the mean $R^2$ score value, and $n$ is the total number of training sessions). The performance results obtained are presented in Table 1.

$$\sigma = \sqrt{\frac{\sum x_i - \overline{x}}{n}} \tag{3}$$

**Table 1.** Mean performance results on the training and test sets using the five selected CNN architectures trained for the arrow orientation prediction task.

| Performance/ Model | Training Set | | | Test Set | | | | Number of Parameters | Mean Training Time (s/Epoch) | Mean Inference Time (s) |
|---|---|---|---|---|---|---|---|---|---|---|
| | Loss | Angular Error | $R^2$ Score | Loss | Angular Error | $R^2$ Score | Stdev. of the $R^2$ Score | | | |
| Ad hoc | 0.0011 | 2.5162 | 0.9874 | 0.0136 | 6.5801 | 0.8440 | 0.0325 | 2,673,729 | 1.63 | 2.59 |
| ResNet-50 | 0.0014 | 3.2250 | 0.9807 | 0.0156 | 8.6915 | 0.8045 | 0.0706 | 24,671,745 | 6.85 | 4.31 |
| VGG16 | 0.0001 | 1.3400 | 0.9984 | 0.0137 | 6.6425 | 0.8320 | 0.0452 | 15,012,289 | 7.40 | 11.97 |
| VGG19 | 0.0006 | 2.1564 | 0.9926 | 0.0111 | 5.5975 | 0.8683 | 0.0419 | 20,321,985 | 8.56 | 14.89 |
| Xception | 0.0006 | 3.0064 | 0.9883 | 0.0173 | 9.9843 | 0.7928 | 0.0487 | 21,945,513 | 11.05 | 4.87 |
| DenseNet-121 | 0.0016 | 8.3155 | 0.7760 | 0.0163 | 10.1833 | 0.7946 | 0.0456 | 8,223,915 | 7.65 | 3.08 |

The results show that the VGG16 and VGG19 variants of VGGNet achieved the best performance, with mean angular errors of 1.34 and 2.16, on the training set, respectively, and $R^2$ scores of 0.87 and 0.83, on the test set, respectively. ResNet-50 and Xception performed slightly worse, with mean angular errors of 3.23 and 3.01, respectively, and lower validation $R^2$ scores of 0.80 and 0.79, respectively. Meanwhile, DenseNet-121 exhibited a relatively higher mean angular error of 10.18, with a test $R^2$ score of 0.79, and a standard deviation of the test $R^2$ score of 0.05.

The proposed ad hoc model displayed a high generalisation capability in predicting the target variable, achieving a mean angular error of 2.52 degrees on the training set and a test $R^2$ score of 0.84. These values are remarkable when considering the model's increased computational efficiency (the ad hoc model processed and predicted the available information from 4.3 times to 6.2 times faster when compared with the other NN candidates). This indicates its appropriateness for use in similar regression tasks. In addition, the ad hoc model was the one with the most consistent performance, as its standard deviation of the test $R^2$ scores reached a minimum of 0.03. Nonetheless, the standard deviations of the test $R^2$ scores were relatively low across all models, reaching a value of 0.05 for DenseNet-121 and a maximum of 0.07 in the case of ResNet-50.

As for the proposed ad hoc model, its training process was up to 6.2 times faster when compared with its well-established counterparts, which indicates an advantage in applications where the real-time detection of arrow orientation is pursued. One possible explanation is that it features fewer layers and parameters when compared with well-established architectures. Moreover, during inference, it consistently performed between 1.2 and 5.7 times more rapidly. This advantage can be significant in real-world applications where real-time detection of arrow orientation is necessary. Such scenarios might include high-speed autonomous vehicles or robotics applications where rapid decision-making is crucial. One possible reason for this speed advantage is that the ad hoc model has fewer layers and parameters than the more established architectures, mitigating the risk of overfitting, which is a common issue in deep learning models with large parameter spaces. This model, therefore, offers a promising solution for applications where speed and efficiency are key factors. However, it is important to mention that the ad hoc model had to learn the studied phenomenon from scratch, which may have influenced its capacity to learn and generalise patterns in the data.

It is also important to note that the loss metric used in our models does not consider the potential error in arrows that are near 0 degrees, causing the error measurement between 0 and 359 degrees to be much larger than it is. However, given the ability of the models

to tolerate noise, this is not a significant concern. Nonetheless, future work could explore alternative loss functions that account for this phenomenon to further improve accuracy. To gain a better understanding of the values presented in Table 1 and provide a clear visual representation of how the models compare to each other, the performance of the trained models is also presented in Figure 10 in terms of the $R^2$ score, MSE, and angular error.

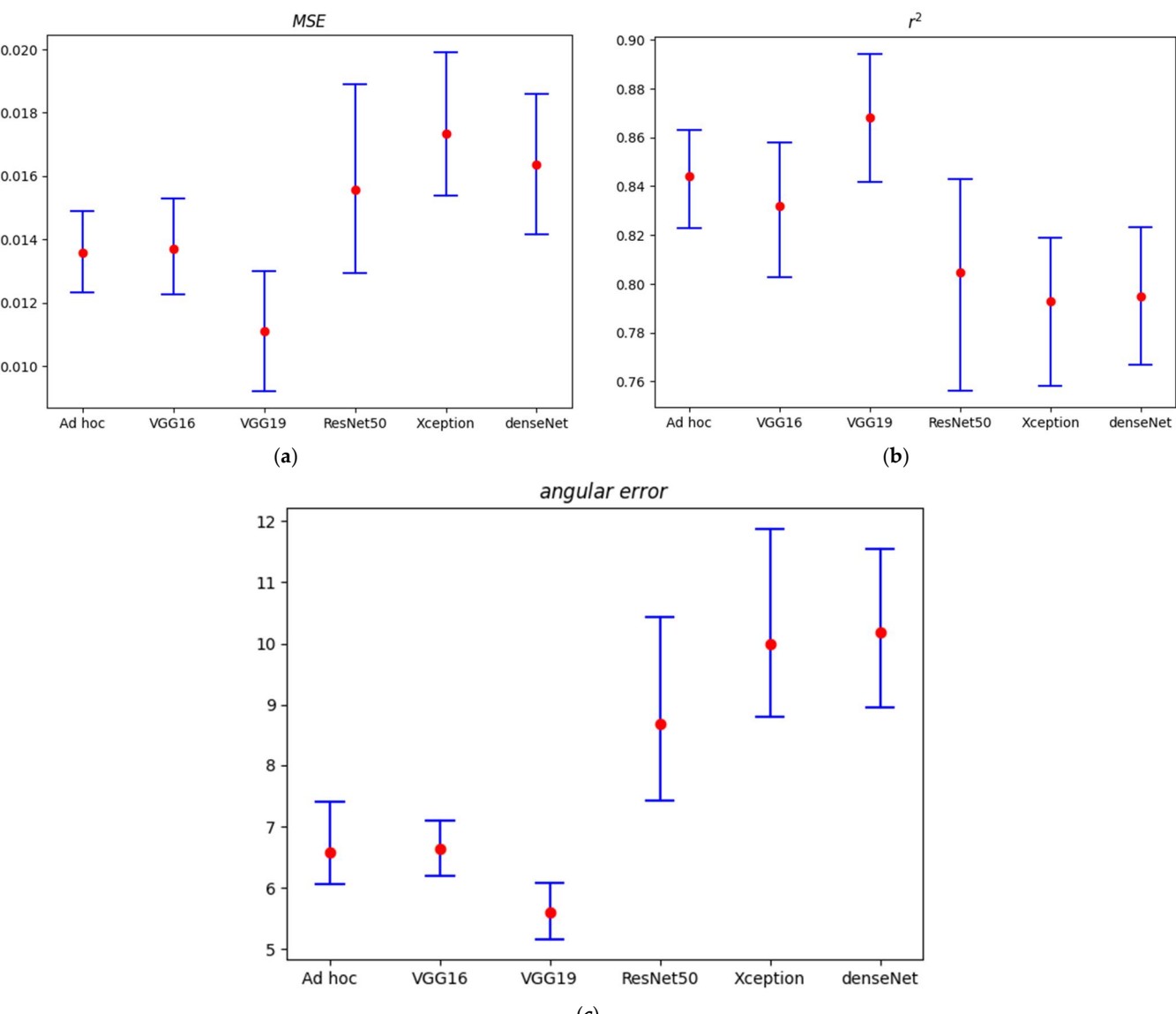

**Figure 10.** Visual representation showing the performance metrics achieved on the test set using the trained models in terms of (**a**) $R^2$ score, (**b**) MSE, and (**c**) angular error. Note: The intervals represent the values obtained from applying the bootstrapping training procedure (described in Section 3.3).

According to these results, the first highlighted aspect is that Xception and ResNet-50, despite generally having a higher feature extraction capability, display a relatively worse predictive performance for this regression task. Interestingly, the results also suggest that more powerful architectures, such as Xception and ResNet-50, although pre-trained on ImageNet, may not always generalise well to other computer vision tasks. Despite their significant performance on ImageNet, VGG16 and VGG19 outperformed both Xception and ResNet50 on our task, as measured using the $R^2$ score and angular error. This suggests that the features learned using these architectures may not be as relevant to our task as in the case of ImageNet. Thus, while pre-trained models featuring many parameters can be a useful starting point for many computer vision tasks, they may not always be the

best choice, and other architectures should be considered depending on the specifics of the problem. Surprisingly, the VGG16 and VGG19 models, despite their slower processing times compared with our ad hoc model, outperformed Xception, ResNet-50, and DenseNet-121 on the approached task. This superior performance could be critically advantageous in applications where the slightest angular error in arrow orientation prediction could lead to significant consequences, such as misrouting in navigation systems.

As for overfitting concerns, the appropriate use of regularisation techniques (specifically, the dropout technique) prevents the model from memorising noise in the training data. In addition, as explained in Sections 3.2 and 4.3, for higher control of overfitting behaviour, data augmentation (changes in brightness and contrast or shifts in gamma and channel intensities) was applied together with the bootstrapping technique for training (so that the division of the dataset and the training / validation were repeated ten times). The results obtained using the train and test sets display $R^2$ scores that approach 0.9, and the boxplots for the performance metrics do not display strong indicators of overfitting behaviour.

Despite the high feature extraction capability of models such as Xception, ResNet-50, and DenseNet-121, the model did not perform well in the approach regression task. The real-world implications of these displayed performances are important, especially in critical applications such as autonomous vehicles or robotics, where even small errors in determining the direction of an arrow could result in significant deviation. It can be highlighted that, although VGG16 and VGG19 are slower, they are more accurate than other models, indicating their potential usefulness in scenarios where the highest accuracy is needed (such as in navigation systems). Explicitly put, the inference speed of the ad hoc model may be important in real-world applications, where real-time detection of arrow orientation is necessary (for example, in high-speed autonomous vehicles, or robotics applications, where fast decision-making is crucial), making the ad hoc model more suitable in cases that demand real-time detection and quick decision-making.

In relation to the uncertainties in the models, the quantitative results listed in Table 1 (especially the standard deviation) and the graphical representation of the performance in the form of boxplots showcasing the distribution of MSE, $R^2$, and angular error metrics (in Figure 10) report a robust overview of the variability and reliability in the predictions of the models.

Regarding the interpretability of the models, the challenges associated with deep learning models are understood. While this work did not use specific techniques for feature interpretation, the ad hoc model architecture was designed with simplicity in mind, favouring transparency over complexity. However, six random test scenarios where the predictions feature high angular errors (more than 30 degrees) are reported in Figure 11. Higher error rates were generally observed in complex scenes, where several linear elements (such as lane separation lines, as illustrated in Figure 11a–c) with similar characteristics are present. Another important source of error is represented by the complex nature of the tackled task, as the studied arrow elements feature reduced dimensions and the corresponding samples display blurry, unclear arrows, even when using the highest available orthoimages with a spatial resolution of 50 cm (as found in Figure 11d,f). Furthermore, obstructions present in the scenes (such as scenes) also seem to have an important impact on the quality of the predictions; considerably higher error rates are encountered in such scenarios (as displayed in Figure 11e).

Regarding the robustness of the model in various additional scenarios, the data used in this study are based on high-resolution orthoimages that were captured by a public agency under optimal lighting conditions. Consequently, the training data aligns with these favourable lighting conditions, but it is expected that the trained models display improved robustness, due to the data augmentation techniques applied to expose the model to a range of lighting scenarios commonly encountered in real-world settings (that include variations in brightness and contrast).

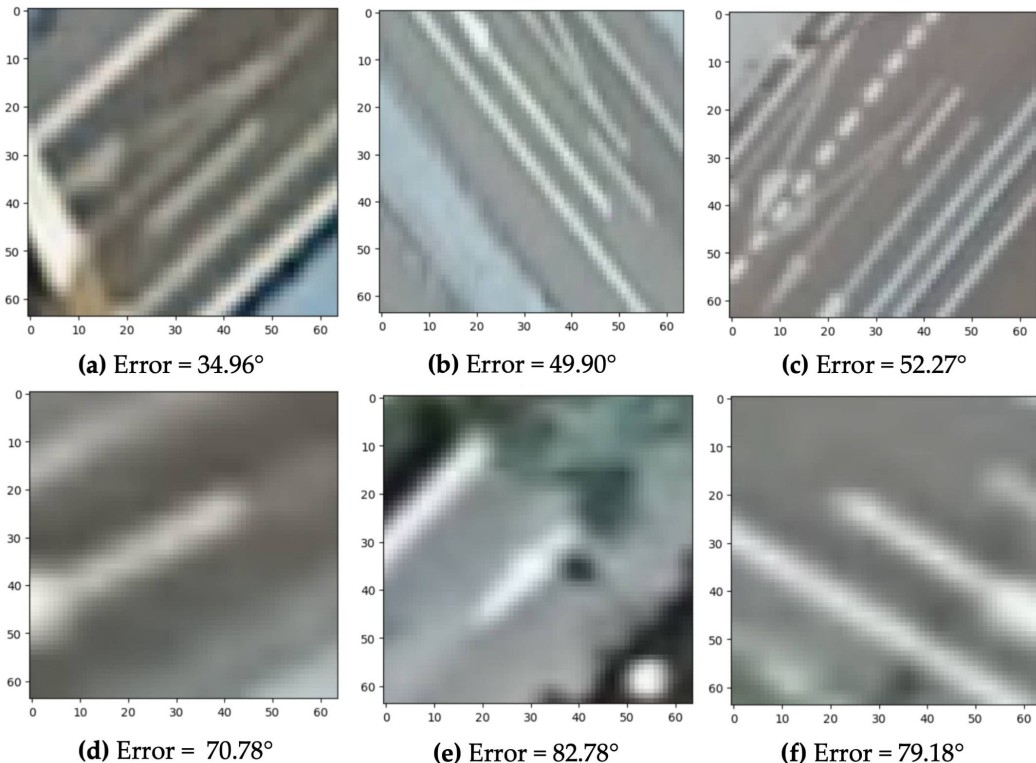

**Figure 11.** (**a**–**f**) Random samples featuring high predicted angular error (superior to 30 degrees) that were obtained using the ad hoc model.

Overall, the decision over the use of a certain model should be dictated by the specific application, the level of accuracy required, and the degree of computational efficiency needed. Future studies should aim to improve the trade-off between these factors for more robust and versatile computer vision tasks. For example, in a real-world setting, such as an autonomous vehicle or robotics application, where determining the direction of an arrow is crucial, the increased error rate in these models could result in a significant misdirection. In particular, DenseNet-121 showed a significantly higher mean angular error of 10.1833, along with a test $R^2$ score of 0.7946, reflecting its poorer performance compared with the other models. This suggests that the features learned using these architectures may not be as suitable for tasks like ours as they are for ImageNet, leading to the potential overfitting to ImageNet. Thus, while these pre-trained models can provide a strong foundation for many computer vision tasks, their application should be carefully considered based on the specifics of the problem at hand.

## 6. Conclusions

The proposed approach has the potential to significantly improve the accuracy and efficiency of road sign identification and ultimately contribute to the development of safer and more efficient transportation systems. The ad hoc model proposed was trained from scratch and delivered a high performance, indicating that it may be possible to develop custom models for specific applications, and it was most consistent in its predictions (lowest standard deviation on the test set).

The results of this study also demonstrate the importance of carefully selecting and evaluating CNN models for specific tasks and suggest that CNN architectures modified for regression tasks can be effective for arrow angle estimation in images. The models based on VGG16 and VGG19, which were pre-trained using a dataset with more than one million images, were able to effectively learn and generalise patterns in the data, achieving the highest performance metrics. However, the achievement of these results might have been greatly incentivised by applying transfer learning techniques for training.

Further research is needed to determine the optimal architecture and training methodology for CNN models in applications based on regression tasks. Future work could also explore the performance of these models on larger and more diverse datasets, as well as investigate the use of ensemble methods for achieving an improved performance.

In addition, a real-world evaluation of the model (in the form of tests to validate the practical utility and applicability of our approach) is expected in the future, due to the resource-intensive process that requires specialised equipment to obtain accurate testing data (aerial orthoimages or image data collected by autonomous vehicles). In parallel, the addition of the predicted data as a traffic direction attribute, once the road axes are identified using semantic segmentation and the traffic direction arrow is identified, will also be explored for real-time use in an on-board driving system.

**Author Contributions:** C.-I.C.: formal analysis, investigation, methodology, validation, visualisation, writing—original draft, and writing—review and editing; A.D.-Á.: conceptualisation, data curation, investigation, methodology, software, validation, visualisation, writing—original draft, and writing—review and editing; F.S.: conceptualisation, data curation, funding acquisition, investigation, methodology, project administration, resources, software, supervision, validation, and writing—review and editing; M.-Á.M.-C.: conceptualisation, data curation, funding acquisition, investigation, methodology, project administration, resources, software, supervision, validation, visualisation, writing—original draft, and writing—review and editing. All authors have read and agreed to the published version of the manuscript.

**Funding:** This research is part of the "Deep learning applied to the recognition, semantic segmentation, post-processing, and extraction of the geometry of main roads, secondary roads and paths (SROADEX)" project (PID2020-116448GB-I00) funded by the AEI (MCIN/AEI/10.13039/501100011033).

**Data Availability Statement:** The RoadArrowORIEN dataset (approximately 6700 images) used for training and testing the model required in the methodology proposed in this manuscript is openly available under a CC-BY 4.0 licence and can be downloaded from the Zenodo data repository using the link: 10.5281/zenodo.7840642.

**Conflicts of Interest:** The authors declare no conflict of interest. The funders had no role in the design of this study; in the collection, analyses, or interpretation of data; in the writing of this manuscript; or in the decision to publish the results.

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
