# Peer review of "Convolutional Neural Networks Adapted for Regression Tasks: Predicting the Orientation of Straight Arrows on Marked Road Pavement Using Deep Learning and Rectified Orthophotography"

_electronics, doi:10.3390/electronics12183980_

Round 1
Reviewer 1 Report
The manuscript entitled “Convolutional Neural Networks Adapted for Regression Tasks: Predicting the Orientation of Straight Arrows on Marked Roads Pavement using Deep Learning and Rectified Orthophotography” has been investigated in detail.
The paper’s subject could be interesting for readers of journal. Therefore, I recommend this paper for publication in this journal but before that, I have a few comments on the text that should be addressed before publication:
· Some abbreviations are missing in the text, please check the manuscript thoroughly to find the ones and add them in the table.
· What is application and importance of proposed method in this investigation in the real life?
· Authors should clearly write what the motivation of this paper is.
· How did you evaluate the accuracy of obtained results?
· The linguistic quality needs improvement. It is essential to make sure that the manuscript reads smoothly- this definitely helps the reader fully appreciate your research findings. There are grammar and writing style errors that should be corrected by the authors.
· Mentioned results of present work in the conclusion section must be compared with the previous studies in the field
· Since recently it has been proved that computational techniques, specifically machine learning has a numerous applications in all of engineering fields, I highly recommend the authors to add some references in this manuscript in this regard. It would be useful for the readers of journal to get familiar with the application of computational techniques in other engineering fields. I recommend the others to add all the following references, which are the newest references in this field
[1] Seitllari, A., Ghazavi, M., & Kutay, M. E. (2020). Effects of binder modification on rutting performance of asphalt binders. In Proceedings of the 9th International Conference on Maintenance and Rehabilitation of Pavements—Mairepav9 (pp. 607-615). Springer, Cham.
Suspensions. Physical Review Letters, 129(6), 068001."
[2] Roshani, et al., 2018. Density and velocity determination for single-phase flow based on radiotracer technique and neural networks. Flow Measurement and Instrumentation, 61, pp.9-14
[3] Mohaidat, M., Grantner, J. L., Shebrain, S. A., & Abdel-Qader, I. (2022, May). Instrument detection for the intracorporeal suturing task in the laparoscopic box trainer using single-stage object detectors. In 2022 IEEE International Conference on Electro Information Technology (eIT) (pp. 455-460). IEEE.
Author Response
The authors thank the reviewer for taking their time to review the article, raise the questions and provide all the comments.
Point#1: Some abbreviations are missing in the text, please check the manuscript thoroughly to find the ones and add them in the table.
Response#1: The manuscript was rechecked for the problem of missing abbreviations and acronyms and the cases identified were corrected.
Point#2: What is application and importance of proposed method in this investigation in the real life?
Response#2: The application for which the method has been investigated and proposed is to be able to automatically determine the direction of travel of lanes of highways or road networks that flow in parallel. To associate it within the scope of the production or updating of cartography that facilitates autonomous vehicle navigation. This information is associated with the geometries of the road axes along with other types of information such as the number of lanes or speed limits. To solve the comment, this aspect was added in the Introduction section (lines 47-53).
Point#3: Authors should clearly write what the motivation of this paper is.
Response#3: No methodological proposal has been found in the literature that directly identifies the orientation of a traffic direction arrow in roadways. The closest that has been found are proposed networks that identify objects in images based on modifications of the YOLO architecture to identify the orientation of the enveloping rectangle of the objects, which enables the recovery of the object’s area, but not the direction of the arrow. In addition, parallel lanes on highways would not be able to differentiate between arrows of one direction and those of the opposite direction. To solve the comment, further explanations were added in the Related Works section (lines 157-170).
Point#4: How did you evaluate the accuracy of obtained results?
Response#4: The accuracy of the obtained results by employing a comprehensive set of evaluation metrics. Specifically, the following metrics to evaluate the effectiveness of the five trained models were utilised: MSE (loss), the R2 score and MAE (as found at the beginning of Section 5, lines 464-470). The detailed results of the model evaluations, including the values of these metrics, are presented in the Results section.
Point#5: The linguistic quality needs improvement. It is essential to make sure that the manuscript reads smoothly- this definitely helps the reader fully appreciate your research findings. There are grammar and writing style errors that should be corrected by the authors.
Response#5: To solve this comment, further efforts were made to improve the English grammar, style and syntax of the manuscript.
Point#6: Mentioned results of present work in the conclusion section must be compared with the previous studies in the field.
Response#6: No similar work has been identified in the literature; the results delivered by the proposed method are not comparable for now. However, the results of using different popular deep convolutional network architectures that were adapted for regression tasks by the proposed technique are compared in Section 4 and 5. For solving this comment, the novelty of the method was reinforced in Sections 1 and 2 (lines 47-53 and lines 157-170).
Point#7: Since recently it has been proved that computational techniques, specifically machine learning has a numerous applications in all of engineering fields, I highly recommend the authors to add some references in this manuscript in this regard. It would be useful for the readers of journal to get familiar with the application of computational techniques in other engineering fields. I recommend the others to add all the following references, which are the newest references in this field:
- [1] Seitllari, A., Ghazavi, M., & Kutay, M. E. (2020). Effects of binder modification on rutting performance of asphalt binders. In Proceedings of the 9th International Conference on Maintenance and Rehabilitation of Pavements—Mairepav9 (pp. 607-615). Springer, Cham. Suspensions. Physical Review Letters, 129(6), 068001."
- [2] Roshani, et al., 2018. Density and velocity determination for single-phase flow based on radiotracer technique and neural networks. Flow Measurement and Instrumentation, 61, pp.9-14
- [3] Mohaidat, M., Grantner, J. L., Shebrain, S. A., & Abdel-Qader, I. (2022, May). Instrument detection for the intracorporeal suturing task in the laparoscopic box trainer using single-stage object detectors. In 2022 IEEE International Conference on Electro Information Technology (eIT) (pp. 455-460). IEEE.
Response#7: The authors appreciate the comment and the proposals, but consider that the mentioned articles have a different scope when compared to this work (which is to determine the angle of the traffic direction arrow using deep learning techniques). In this regard, the first article deals with asphalt manufacturing components (binders). The second one, deals with the measurement of density and velocity of fluids (and not with the pavement arrow direction). The third one deals with object detection with YOLO networks in the hospital environment to monitor suturing tasks. Therefore, the study objectives of these papers are not related with the determination of the direction or angle of objects found on pavement and were not included as citations.
Reviewer 2 Report
The proposed study presents an interesting deep learning-based approach for identifying the orientation and direction of arrow signs on marked roadway pavements. This is certainly a vital task, considering the rise of autonomous vehicles. However, there are several aspects that could be improved or clarified:
1. There is no mention of the dataset used in this study. How many images were used? How were they labeled? The quality and quantity of the data used would significantly impact the model's performance and its potential to generalize to different situations and environments.
2. How was the validation performed? Details about the train-test split and cross-validation strategy should be provided. Furthermore, considering the high accuracy of the models (R² score of 0.9874 for the ad hoc model), it might be worth investigating whether there was any leakage of information from the training set to the test set, which could lead to overly optimistic results.
3. Although the ad hoc model's performance was compared with the VGG-19 model, it would be beneficial to compare the performance with other state-of-the-art models for image recognition tasks (such as ResNet, DenseNet, etc.) to understand how well the ad hoc model performs in relation to other established models.
4. Given the complex nature of deep learning models and their "black-box" criticism, it might be beneficial to discuss the interpretability of the models. Furthermore, with such a high R² score, it's crucial to ensure that the model is not overfitting the data.
5. Although the ad hoc model is computationally more efficient than the VGG-19 model, it's important to consider how this efficiency is measured. If it's purely based on the number of parameters, it may not reflect the actual computational cost during training and inference, especially if different layers have different computational costs.
6. Robustness of the model in various scenarios is another important aspect that should be considered. Does the model perform well in different lighting conditions, weather conditions, or when the arrow signs are partially occluded?
7. CNN is well-known and has been used in previous studies i.e., PMID: 36166351, PMID: 36642410. Therefore, the authors are suggested to refer to more works in this description to attract a broader readership.
8. It would be beneficial to explain how the model would be integrated into an autonomous driving system and how it would interact with other components of such a system.
9. Finally, it would be helpful to see a real-world evaluation of the model. This could include applying the model to new aerial orthoimages or even images collected by autonomous vehicles themselves.
10. There are 2 sections "Related Work".
11. Uncertainties of models should be reported.
12. More discussions should be added.
English writing and presentation style should be improved.
Author Response
The authors thank the reviewer for taking their time to review the article, raise the questions and provide the comments.
Point#1: There is no mention of the dataset used in this study. How many images were used? How were they labeled? The quality and quantity of the data used would significantly impact the model's performance and its potential to generalize to different situations and environments.
Response#1: As found in the Data Availability Statement and throughout the paper, the dataset used in this work is RoadArrowORIEN: dataset of 6701 images (64x64 pixels) of straight arrow-type road markings and their azimuths 2023 (Manso Callejo, M.Á.; García, F.S.; Cira, C.-I.), openly available at https://zenodo.org/record/7840642).
As described in Sections 3.1 and 4.1, RoadArrowORIEN was generated by manually tagging straight arrows found in aerial orthoimages with LabelMe and following the automatic process presented in Figure 2, together with a subsequent validation by a specialised operator. Regarding the data quality, and as specified in Section 4.1, it consists of 6701 64x64-pixels aerial images with a spatial resolution of 25 cm that were provided and corrected by a public agency, the Spanish National Geographical Institute. The procedure is additionally described in the Zenodo repository. The authors consider that the raised aspects are extensively and sufficiently described and presented in the manuscript.
Point#2. How was the validation performed? Details about the train-test split and cross-validation strategy should be provided. Furthermore, considering the high accuracy of the models (R² score of 0.9874 for the ad hoc model), it might be worth investigating whether there was any leakage of information from the training set to the test set, which could lead to overly optimistic results.
Response#2: Lines 445-446 indicate how the dataset was randomly split by applying the 90-10% criterion for obtaining the training and test sets. In addition, as illustrated in Phase 2 of Figure 1, the random division of the dataset for training and validation involved Bootstrapping in the training, so that the division of the dataset and the training/validation has been repeated ten times in order to reduce the variance and avoid overfitting as is explained in section 3.3. For solving this comment, further explanations were also added in lines 446-452.
Point#3. Although the ad hoc model's performance was compared with the VGG-19 model, it would be beneficial to compare the performance with other state-of-the-art models for image recognition tasks (such as ResNet, DenseNet, etc.) to understand how well the ad hoc model performs in relation to other established models.
Response#3: The results delivered by the Ad hoc model were compared to those obtained by the regression adaptations of VGG16 and VGG19 and ResNet-50 and Xception in the first version of the paper.
Following the reviewer's suggestion, an adapted version of the DenseNet model (with the regression structure proposed in this paper) has been included in the benchmark (as can be seen in Table 1 of the current version of the paper, and the the newly created subsection 4.2.4.). In this regard, the Discussion section was also extended to analyse the performance from different perspectives; the analysis of multiple models was reinforced throughout the paper (for example, in the Abstract and Introduction sections).
Point#4. Given the complex nature of deep learning models and their "black-box" criticism, it might be beneficial to discuss the interpretability of the models. Furthermore, with such a high R² score, it's crucial to ensure that the model is not overfitting the data.
Response#4: Regarding model interpretability, we understand the challenges associated with deep learning models. While we did not employ specific techniques for feature interpretation, we designed our model architecture with simplicity in mind, favouring transparency over complexity. This aspect regarding the model interpretability was commented in the Discussion section (lines 566-569).
As for overfitting concerns, the appropriate use of regularisation techniques (spe-cifically dropout technique) prevent the model from memorising noise in the training data. In addition, as explained in Sections 3.2 and 4.3, for a higher control of the overfitting behaviour, data augmentation (changes in brightness and contrast or shifts in gamma and channel intensities) were applied together with the bootstrapping technique for training (so that the division of the dataset and the training / validation has been repeated ten times). The results obtained on train and test sets display R2 scores that approach 0.9 and the boxplots of the performance metrics do not display strong indicators of the overfitting behaviour. These aspects were reinforced in the Discussion sections (lines 539-549), and commented in Section 3.2 and 4.3.
Point#5. Although the ad hoc model is computationally more efficient than the VGG-19 model, it's important to consider how this efficiency is measured. If it's purely based on the number of parameters, it may not reflect the actual computational cost during training and inference, especially if different layers have different computational costs.
Response#5: In response to this observation, the paper has been extensively modified to include a more comprehensive explanation of how the computational efficiency was evaluated. Specifically, the 'Result and Discussion' section of the paper has been expanded with more information (line 491-505). We hope that these additions provide a clearer understanding of the practical computational cost considerations in this study, addressing your concern about the measurement of the computational efficiency, while illustrating the advantages of our ad hoc model in real-world applications.
Point#6. Robustness of the model in various scenarios is another important aspect that should be considered. Does the model perform well in different lighting conditions, weather conditions, or when the arrow signs are partially occluded?
Response#6: The primary focus in this study was on the use of high-resolution orthoimages, which are captured under optimal lighting conditions. These images are obtained every 2 to 4 years and are typically taken during the summer when lighting conditions are consistent. Consequently, our training data aligns with these favourable lighting conditions. However, to further assess the model's robustness, we have applied data augmentation techniques, including variations in brightness and contrast, which expose the model to a range of lighting scenarios commonly encountered in real-world settings. This aspect was also commented in 4.1. Data (lines 345-354) and 5. Results and Discussion sections (lines 583-589).
Point#7. CNN is well-known and has been used in previous studies i.e., PMID: 36166351, PMID: 36642410. Therefore, the authors are suggested to refer to more works in this description to attract a broader readership.
Response#7: The two patents identified are also related to PMID: 36642410 "Prediction of anticancer peptides based on an ensemble model of deep learning and machine learning using ordinal positional encoding" and PMID: 36166351 "Identifying SNARE Proteins Using an Alignment-Free Method Based on Multiscan Convolutional Neural Network and PSSM Profiles". The authors consider that the tasks to be solved in the mentioned studies are not similar. In those cases, the objectives of the studies are related to identifying objects (classification) and are not tackling regression tasks.
The authors welcome suggestions, but that the objective of our study was to determine the angle of the traffic direction arrow using deep learning techniques (a regression taks). To solve this comment, we reinforced the novelty of this study in lines (lines 47-53 and lines 157-170).
Point#8. It would be beneficial to explain how the model would be integrated into an autonomous driving system and how it would interact with other components of such a system.
Response#8: This is indeed a very interesting suggestion, but it falls outside the scope of this work. The aim of this paper is to identify the direction arrows on the roads in aerial orthophotography and to complement them with the angle or direction of the traffic on the roadway, and not to use them in real time in an ADA on-board system. For solving this comment, the mentioned aspect was included as a possible future line of research in lines 628-634 (the addition of the predicted data as a traffic direction attribute, once the road axes have been identified by semantic segmentation and the traffic direction arrow has been identified).
Point#9. Finally, it would be helpful to see a real-world evaluation of the model. This could include applying the model to new aerial orthoimages or even images collected by autonomous vehicles themselves.
Response#9: We recognize the importance of such tests to validate the practical utility and applicability of our approach. We have added lines 573-586 and Figure 12 (where we present scenes with high error rates) to discuss and present a sample-based evaluation of the test results. At the present moment, we currently do not have access to an additional dataset of ortho rectified aerial images as obtaining high-quality and precise images is a resource-intensive process that requires specialised equipment and expertise. Given these challenges, while we have not yet conducted such real-world tests, we remain hopeful to explore this line in the future. To solve this comment, we have added lines 566-579 and Figure 12 and a mentioning of this aspect as a future line of research (lines 622-628).
Point#10. There are 2 sections "Related Work".
Response#10: The mistake was corrected and Section 3 was renamed to "Method Proposal".
Point#11. Uncertainties of models should be reported.
Response#11: We authors thank the reviewer for highlighting the importance of addressing model uncertainties. We have included both quantitative and graphical representations to report these uncertainties. Specifically, the standard deviation of the test R² scores has been provided for each model in the 'Results and Discussion' section. Additionally, boxplots showcasing the distribution of MSE, R², and angular error metrics have been added. Moreover, a comprehensive table enumerating metrics such as train/test loss, angular error, and R² score for each model is also presented. We believe these additions offer a robust overview of the variability and reliability of the models' predictions. Further explanations were added in lines 560-564 and throughout the Discussion section.
Point#12. More discussions should be added.
Response#12: For solving this comment, as well as feedback from other reviewers, a comprehensive revision of the "Results and discussion" section has been undertaken. As it can be found in the current version of the manuscript, this section has been expanded to provide a more in-depth analysis, incorporating additional interpretations, comparisons, and potential implications of our findings.
Reviewer 3 Report
The paper proposes a deep learning-based approach for identifying arrow sign orientation on roadway pavements. However, several key concerns must be addressed to strengthen the work and its contributions.
Firstly, the introduction should include a thorough literature review and compare the proposed approach with existing methods. This context is essential for understanding the novelty and significance of the proposed approach in the field.
Additionally, while the paper mentions that the best-performing neural network is based on VGGNet, it lacks a comparative analysis with other state-of-the-art models. A comprehensive evaluation with different relevant neural network architectures would better understand the model's performance and superiority over existing alternatives.
Furthermore, the ad hoc model's promising results require a more detailed description and analysis of its architecture to ensure reliability and generalizability. Providing in-depth information about the design and rationale behind the ad hoc model would justify its effectiveness.
The evaluation metrics used for comparing the models are appropriate, but the paper should discuss the practical implications of the results. Understanding how accurately the models predict arrow sign orientation and direction in real-world scenarios is crucial. The significance of the slight performance differences between the VGGNet and the ad hoc model needs to be addressed.
A more comprehensive analysis of both models' computational requirements, including inference time and memory usage, would clarify the trade-offs between accuracy and efficiency.
Author Response
The authors would like to thank the reviewer for their feedback.
Point#1: Firstly, the introduction should include a thorough literature review and compare the proposed approach with existing methods. This context is essential for understanding the novelty and significance of the proposed approach in the field.
Response#1: The authors understand that many works feature the literature review in the “Introduction” section. However, in the case of this manuscript, joining the two sections would greatly difficult the reading due to excessive length. For this reason, a Section 2 of “Related Works” was created, where existing works are described and discussed independently, as it is generally recommended that introductions should not get into specifics and focus on briefly describing the existing problem, a summary of the solution and a point-to-point summary of the contributions.
Point#2: Additionally, while the paper mentions that the best-performing neural network is based on VGGNet, it lacks a comparative analysis with other state-of-the-art models. A comprehensive evaluation with different relevant neural network architectures would better understand the model's performance and superiority over existing alternatives.
Response#2: The authors understand the importance of benchmarking our approach against a variety of popular architectures in order to understand its relative performance and potential advantages. In the initial version of our paper, comparisons with VGGNet, ResNet-50, and Xception were carried out, as these models are considered important reference architectures within the field. In response to this comment, the comparative analysis was expanded to also include DenseNet in the evaluations and analysis (as found in Sections 4.2.3 and 5) due to its popularity and use in several other works. Is it expected that this additional comparison will provide a more comprehensive overview of the proposed method’s performance.
Point#3: Furthermore, the ad hoc model's promising results require a more detailed description and analysis of its architecture to ensure reliability and generalizability. Providing in-depth information about the design and rationale behind the ad hoc model would justify its effectiveness.
Response#3: For solving this comment, Section 3.2 “Proposal of CNN Adaptation for Regression Tasks and Ad Hoc Model Architecture" has been substantially updated to delve deeper into the design, rationale, and architectural intricacies of the proposed ad hoc model (as found in newer version of the document, lines 252-301).
Point#4: The evaluation metrics used for comparing the models are appropriate, but the paper should discuss the practical implications of the results. Understanding how accurately the models predict arrow sign orientation and direction in real-world scenarios is crucial. The significance of the slight performance differences between the VGGNet and the ad hoc model needs to be addressed.
Response#4: To address the mentioned concerns, the “Results and discussion” section has been revised to provide more context on the implications of our findings in real-world scenarios. We elaborated on the potential impact of the performance differences between the VGG16, VGG19, and the ad hoc model in scenarios such as autonomous vehicle navigation or signal processing in robotics. In lines 548-560 and 590-602, the importance of accuracy versus processing time depending on the specific application was further discussed, together with how our ad hoc model might be more suitable in cases that demand real-time detection and quick decision-making.
Point#5: A more comprehensive analysis of both models' computational requirements, including inference time and memory usage, would clarify the trade-offs between accuracy and efficiency.
Response#5: For solving this comment, the time required for the tile evaluation with the trained models (on the CPU, instead of a dedicated GPU) was measured and the results are reported in Table 1. The discussion section was also enriched to reflect these updates.
Round 2
Reviewer 1 Report
all comments have been addressed correctly and the paper is ready for publication in the present form